# NSCLC in the Era of Targeted and Immunotherapy: What Every Pulmonologist Must Know

**DOI:** 10.3390/diagnostics13061117

**Published:** 2023-03-15

**Authors:** Carley L. Mitchell, Annie L. Zhang, Debora S. Bruno, Francisco A. Almeida

**Affiliations:** 1Department of Internal Medicine, Case Western Reserve University Hospitals, Cleveland, OH 44106, USA; 2Division of Hematology and Oncology, Case Western Reserve University Hospitals/Seidman Cancer Center, Cleveland, OH 44106, USA; 3Department of Pulmonary Medicine, Respiratory Institute, Cleveland Clinic, Cleveland, OH 44195, USA

**Keywords:** non-small cell lung cancer, biomarker testing, targeted therapy, immunotherapy, endobronchial ultrasound, guided bronchoscopy

## Abstract

The treatment of non-small cell lung cancer has dramatically changed over the last decade through the use of targeted therapies and immunotherapies. Implementation of these treatment regimens relies on detailed knowledge regarding each tumor’s specific genomic profile, underscoring the necessity of obtaining superior diagnostic tissue specimens. While these treatment approaches are commonly utilized in the metastatic setting, approval among earlier-stage disease will continue to rise, highlighting the importance of early and comprehensive biomarker testing at the time of diagnosis for all patients. Pulmonologists play an integral role in the diagnosis and staging of non-small cell lung cancer via sophisticated tissue sampling techniques. This multifaceted review will highlight current indications for the use of targeted therapies and immunotherapies in non-small cell lung cancer and will outline the quality of various diagnostic approaches and subsequent success of tissue biomarker testing. Pulmonologist-specific methods, including endobronchial ultrasound and guided bronchoscopy, will be examined as well as other modalities such as CT-guided transthoracic biopsy and more.

## 1. Introduction

Despite a consistent decline in incidence and mortality over the past two decades, lung cancer remains the leading cause of cancer-related mortality for both men and women in the United States (US) and worldwide [1,2]. Non-small cell lung cancer (NSCLC) accounts for approximately 82% of all lung cancer diagnoses [3]. Unfortunately, a dismal 26% five-year overall survival (OS) rate continues to plague this disease [4]. Consequently, efforts to improve diagnosis and treatment remain at the forefront of cancer research. Precision medicine has recently revolutionized the NSCLC treatment landscape, offering benefits including improved progression-free survival (PFS) and OS through the use of targeted therapies and immunotherapies.

Per National Comprehensive Cancer Network (NCCN) guidelines, targeted therapies are now the preferred, first-line agents for metastatic NSCLC among tumors harboring associated oncogenic driver mutations [5]. Despite limited approval in earlier stage disease, ongoing research will continue to transform this space [6]. NCCN guidelines highlight immunotherapy indications for metastatic and locally advanced NSCLC along with resectable disease in the adjuvant and neoadjuvant settings [7]. Ensuring all patients have access to appropriate, individualized therapies requires comprehensive testing at the time of diagnosis. A united multidisciplinary approach among all involved physicians is imperative to achieve this goal.

Pulmonologists are at the forefront of lung cancer diagnoses and play an important role when obtaining the initial biopsy and determining disease staging. Without early and adequate tissue specimens acceptable for histopathologic analysis, immunohistochemistry (IHC), and genomic sequencing, providing appropriate treatment recommendations is impaired. This multifaceted review will highlight current indications for targeted therapies and immunotherapies in NSCLC and the important role of biomarkers and IHC analysis. Additionally, various diagnostic modalities will be explored, outlining the quality of each individual approach.

## 2. Indications for Targeted Therapy

Evolution of targeted therapies is a direct result of improved knowledge regarding the biologic basis of tumor-specific genomic alterations and development of sophisticated means for detecting such abnormalities. Currently, many NSCLC oncogenic driver mutations are recognized with respective targeted drug therapies. These biomarkers include *EGFR* (epidermal growth factor receptor) mutations, *ALK* (anaplastic lymphoma kinase) rearrangements, *ROS*1 (c-ros oncogene 1) rearrangements, *BRAF* (v-raf murine sarcoma viral oncogene homolog B1 mutations, *NTRK* (neurotrophic tyrosine receptor kinase) gene fusions, *MET* (mesenchymal-epithelial transition factor) mutations/amplifications, *RET* (rearranged during transfection) rearrangements, *ERBB2/HER2* (erythroblastic oncogene B/human epidermal growth factor 2 mutations/amplifications, and *KRAS* (kirsten rat sarcoma viral oncogene homolog) mutations. Not only are associated targeted therapies highly effective, with impressive response rates, but they also possess favorable side effect profiles compared to cytotoxic chemotherapy. Those targeting *EGFR* and *ALK* alterations are the most notable and widely studied. Other oncogenic driver mutations and associated therapies are outlined in Table 1/Figure 1.

### 2.1. EGFR-Mutant NSCLC

In 2004, *EGFR* mutations were first described among NSCLC [10], with exon 19 deletions and exon 21 L858R point mutations promoting tumorigenesis in approximately 20% of all lung cancers of adenocarcinoma histology [8]. In 2013, the US Food and Drug Administration (FDA) approved the first *EGFR*-tyrosine kinase inhibitor (TKI), erlotinib, in the first-line setting for *EGFR*-mutant, metastatic NSCLC. The phase III EURTAC trial demonstrated a median PFS benefit of 4.5 months compared to platinum doublet chemotherapy (HR 0.37; 95% CI 0.25–0.54; *p* < 0.0001) along with improved tolerability [10,11] Second-generation *EGFR*-TKIs then emerged followed by the third-generation agent, osimertinib. While active against common *EGFR*-sensitizing mutations, osimertinib is also uniquely selective for the T790M mutation, a common resistance mechanism to the first- and second-generation *EGFR*-TKIs [12]. In fact, osimertinib was first approved in 2015, in the second-line setting, for patients whose tumors developed T790M-positive resistance to earlier agents [12]. Later, the efficacy of osimertinib in previously untreated, metastatic NSCLC was evaluated against first-generation *EGFR*-TKIs in the phase III FLAURA trial. Patients treated with osimertinib derived a 54% reduction in the risk of disease progression or death (HR 0.46; 95% CI 0.37–0.57; *p* < 0.0001), with a median OS of 38.6 months [13,14]. Furthermore, osimertinib displayed superior penetration of the central nervous system (CNS) with significantly greater PFS among this subgroup [13]. This subsequently revolutionized the treatment of such patients, as many with CNS metastases may forego brain radiotherapy until much later in their disease course. Given the impressive results in overall efficacy and tolerability, in 2018, oral osimertinib became, and currently remains, standard of care (SOC) for stage IV, *EGFR*-mutant (ex19del, L858R) NSCLC.

Recently, *EGFR*-TKIs have also led the way for implementation of targeted therapies in early-stage, resectable disease. In the phase III ADAURA trial, adjuvant osimertinib was compared to placebo among *EGFR*-mutant, stage IB-IIIA NSCLC patients following complete resection. Although not required, 60% of patients within each treatment arm received adjuvant chemotherapy, as SOC, prior to enrollment [15]. Osimertinib increased median disease-free survival by 7.2 months for those with stage II-IIIA disease, with 89% of the osimertinib and 59% of the placebo cohorts still alive and disease free at 24 months (HR 0.20; 99.12% CI 0.14–0.30; *p* < 0.001) [15]. Additionally, osimertinib decreased the risk of CNS disease recurrence or death by 82% at 24 months [15]. Outcomes of this trial led to the FDA approval of adjuvant osimertinib, and since 2021, the NCCN has recommended up to three years of adjuvant osimertinib, with or without adjuvant chemotherapy, for resected stage IIB-IIIA or high risk stage IB-IIA, *EGFR*-mutant NSCLC [16]. Although treatment with *EGFR*-TKIs is not yet established in the neoadjuvant or stage IIIB/C settings, the ongoing, respective NeoADAURA and LAURA trials may provide new future direction for this innovative class of targeted therapy [6].

### 2.2. ALK-Rearranged NSCLC

*ALK* rearrangements were first described in 2007. Crizotinib, targeting multiple alterations, including *ALK*, *MET,* and *ROS*1, received accelerated FDA approval in 2011 for *ALK*-rearranged NSCLC due to impressive results from phase I and II trials [17]. Phase III studies followed, including PROFILE 1014, comparing crizotinib to platinum doublet chemotherapy in the first-line setting for metastatic NSCLC harboring *ALK* rearrangements. Oral crizotinib decreased risk of disease progression or death by 55% compared to standard chemotherapy (HR 0.45; CI 0.35–0.60; *p* < 0.001) with additional benefits observed for objective response rate, duration of response, one-year survival, and patient-reported global quality of life [18]. Since then, multiple agents have been approved including alectinib, the first next-generation agent compared to crizotinib in the phase III ALEX trial. Alectinib showed a remarkable median PFS advantage of 23.9 months (HR 0.43; CI 0.32–0.58; *p* < 0.001) [19], as well as superior CNS disease control compared to crizotinib with an 11.8-month median CNS response duration advantage (HR 0.16, CI 0.10–0.28; *p* < 0.001) [20]. While most recently published OS data remains immature, the five-year OS rate for the alectinib and crizotinib cohorts were 62.5% and 45.5%, respectively [19]. Following this trial, in 2017, the FDA approved first-line alectinib for metastatic, *ALK*-rearranged NSCLC. While *ALK*-TKIs have not been approved for early-stage disease, the ongoing, phase III ALCHEMIST-ALK and ALINA trials are evaluating crizotinib and alectinib, respectively, in the adjuvant setting, along with the phase II ALNEO trial that is evaluating alectinib in the neoadjuvant setting [12].

## 3. Indications for Immunotherapy

Tumors evade detection by the host immune system through immune checkpoint pathways, most notably the programmed cell death protein-1 (PD-1)/programmed death-ligand 1 (PD-L1) axis. Development of highly selective antibodies targeting this pathway has also revolutionized the treatment of NSCLC. Approximately 25% of NSCLC patients have high tumor expression of PD-L1, defined as a tumor proportion score (TPS) greater than 50% [21,22] as measured by IHC. Tumors with high PD-L1 expression have a greater likelihood of response to immune checkpoint inhibitors (ICIs) [21,22].

In 2016, the landmark phase III clinical trial, KEYNOTE-024, established immunotherapy as the first-line treatment for metastatic NSCLC with high PD-L1 expression without concomitant *EGFR* mutations or *ALK* rearrangements. Patients treated with pembrolizumab monotherapy had an improved median PFS by 4.3 months compared with the chemotherapy group (HR 0.50; 95% CI 0.37–0.68; *p* < 0.001) [23]. Likewise, median OS improved by 15.8 months among the pembrolizumab arm (HR 0.63; 95% CI 0.47–0.86) [24]. Adverse events occurred less frequently in those treated with pembrolizumab and the rates of serious treatment-related adverse events were similar between the two groups [23,24].

Since promising results from KEYNOTE-024, immunotherapy has been studied in earlier stage disease. The phase III PACIFIC trial investigated the use of durvalumab, an anti-PD-L1 antibody, as consolidative therapy following definitive chemoradiation compared to placebo in patients with nonoperable stage III NSCLC. Median PFS among the durvalumab arm improved by 11.2 months compared to placebo (HR 0.52; 95% CI 0.42–0.65; *p* < 0.001) [25]. The most recent predicted five-year OS rate for patients treated with durvalumab and placebo is 42.9% and 33.4%, respectively [26]. Of note, a subgroup analysis of patients with *EGFR* mutations showed no benefit from durvalumab [25], underscoring the need for comprehensive biomarker testing among patients diagnosed with locally advanced NSCLC.

The next frontier of immunotherapy research in NSCLC is in the perioperative period for early-stage disease. In 2022, initial results from CheckMate816 showed the addition of PD-1 inhibitor, nivolumab, to neoadjuvant platinum-based chemotherapy performed better than neoadjuvant chemotherapy alone for stage IB-IIIA, resectable NSCLC. The primary end-point, median event-free survival for nivolumab plus chemotherapy reached 31.6 months versus 20.8 months for chemotherapy alone (HR 0.63; CI 0.43–0.91; *p* = 0.005) [27]. Furthermore, pathologic response, correlating with long-term benefits of neoadjuvant therapy, was also evaluated. Impressively, 24% of patients treated with the addition of nivolumab achieved a complete pathologic response upon surgical resection, compared to only 2.2% of patients treated solely with chemotherapy [27]. While long-term outcome data remains to be seen, results from CheckMate816 have led to the approval of nivolumab in the neoadjuvant setting [27]. In the adjuvant, phase III IMpower010 trial, the subsequent addition of PD-L1 inhibitor, atezolizumab, to adjuvant platinum-based chemotherapy demonstrated a 34% disease-free survival advantage for patients with stage II-IIIA, PD-L1-positive NSCLC who underwent surgical resection [28]. Hence, for appropriate patients who do not undergo neoadjuvant chemoimmunotherapy, atezolizumab is approved in the adjuvant setting [7]. Several other phase III trials are currently undergoing accrual to assess chemotherapy plus adjuvant PD-L1 blockade in resectable NSCLC [29].

## 4. Indications for Biomarker Testing

As evidenced above, appropriate biomarker and PD-L1 testing are imperative to provide optimal treatment regimens for patients with NSCLC. Comprehensive testing should be implemented regardless of tumor histology. Traditionally, testing all nonsquamous histologies was advised; however, squamous cell carcinomas should also be considered. Approximately 5–10% of these tumors will harbor a targetable mutation; furthermore, tumor heterogeneity cannot be ruled out [16]. While individual tumor histology (e.g., adenocarcinoma or squamous cell carcinoma) should not be the factor to preclude or promote biomarker testing, it is particularly helpful to guide systemic therapy in addition to immunotherapy in the absence of genomic alterations, as the effectiveness of different cytotoxic chemotherapy agents may be affected by intrinsic mechanisms of resistance inherent to different subtypes of NSCLC [30]. In patients with metastatic NSCLC with no actionable genomic alterations, low PD-L1 expression and/or high tumor burden cytotoxic chemotherapy remain the cornerstone of systemic therapy. Moreover, FDA-approved regimens containing the anti-angiogenic agent bevacizumab are contraindicated in patients with squamous NSCLC in view of an increased risk of high rates of life-threatening hemoptysis [31].

When considering disease stage, the NCCN recommends reflex PD-L1 IHC analysis for all locally advanced and metastatic disease. Additionally, patients with stage II-IIIA NSCLC who undergo surgical resection and are candidates for adjuvant chemotherapy should also be tested for PD-L1 to determine eligibility for adjuvant atezolizumab. Similarly, genomic testing is recommended for all metastatic disease, and at a minimum, testing for *EGFR* mutations is recommended among stage I–III disease [7]. Arguably, comprehensive genomic and PD-L1 testing should be implemented, regardless of stage, immediately at the time of diagnosis. Frequently, tissue biopsy is obtained prior to staging confirmation, and automatic reflex testing can expeditiously guide therapy once definitive stage is determined. A pragmatic example is the patient with early-stage disease who may be a candidate for neoadjuvant chemoimmunotherapy, provided the tumor does not harbor *EGFR* or *ALK* alterations. Additionally, many patients with early and locally advanced disease will progress, necessitating this tumor-specific information. If previously obtained, this prevents risks associated with additional invasive procedures and saves valuable time, money, and resources at an important treatment juncture. Furthermore, ongoing clinical trials continue to evaluate the efficacy of numerous targeted therapies in the adjuvant and neoadjuvant arenas, which will likely showcase promising results leading to new FDA approvals. Ultimately, understanding a tumor’s genomic profile can expedite the implementation of late-breaking effective therapies.

Finally, not only is comprehensive genomic testing important when utilizing SOC treatments, but it is equally necessary when considering clinical trial participation, with enrollment eligibility often hinging upon these results. Unsurprisingly, trials involving targeted therapies require this information, as do those involving immunotherapies, as certain genomic alterations are known to augment ICI effectiveness. Assuming each patient is a clinical trial candidate at the time of diagnosis will save time and, more importantly, the risk of an additional future procedure.

## 5. Disparities in Biomarker Testing

While biomarker testing directly informs the selection of treatment options for NSCLC, research shows next generation sequencing (NGS) is underutilized in everyday practice. A review from a real-world electronic health record database showed that NGS-based testing was only observed among 48.7% of all patients with advanced/metastatic NSCLC (n = 14,768) and only 35.5% of patients with nonsquamous NSCLC had NGS-based testing prior to initiation of first-line therapy [32]. Moreover, notable racial disparities were found. For advanced/metastatic NSCLC, black/African-American patients were less likely to undergo biomarker testing at any point during their treatment trajectories (73.6% vs. 76.4%, *p* = 0.03), especially prior to first-line therapy (25.8% vs. 31.5%; *p* < 0.0001), compared to their white counterparts [32]. As biomarker testing has become standard, follow-up research is needed to ensure all patients receive equitable care.

## 6. Tissue Specimen & Diagnostic Testing

Optimal treatment of lung cancer is determined by biopsy results and staging. If there is clinical evidence of metastasis, sampling from distant sites should be prioritized to confirm stage IV disease. Staging scans can help identify clinical nodal disease, however, it is important to obtain pathologic confirmation, ensuring accurate staging. For example, a retrospective study found a stage shift in nearly 20% of patients who underwent endobronchial ultrasound (EBUS) compared to PET-CT alone [33]. Furthermore, accurate mediastinal staging affects treatment planning for both surgical and nonsurgical patients. It is important to differentiate mediastinal N0/N1 disease from N2/N3 disease, as this influences surgical candidacy. Similarly, nodal involvement may also impact radiation treatment planning. There are many diagnostic approaches available; however, only EBUS-transbronchial needle aspiration (TBNA) and guided bronchoscopies allow concurrent hilar and mediastinal node sampling, making them prime modalities for NSCLC diagnosis and staging. A comprehensive flow chart outlining the suggested tissue diagnosis modalities and subsequent biomarker testing is provided in Figure 2.

### 6.1. Endobronchial Ultrasound

EBUS utilizes an ultrasound transducer located on the tip of a flexible bronchoscope allowing visualization of structures beyond the airway wall and real-time disease sampling. EBUS-TBNA advantageously provides access to a wide array of lymph node stations (2R, 2L, 3p, 4R, 4L, 7, 10R, 10L, 11s, 11i, 11L, 12R, 12L). A method for nodal identification using EBUS and proper biopsy order has previously been described [34,35]. In some cases, sampling of a centrally located primary parenchymal lesion is also possible [36]. The median diagnostic sensitivity of EBUS-TBNA is reportedly 81–89% [35], with low complication risks.

Several studies have demonstrated the adequacy of validated techniques for the use of smears in IHC, NGS, and liquid-based cytology preparations [37,38,39,40]. A meta-analysis of 33 studies reported a pooled probability of adequate material for molecular analysis from EBUS-TBNA sampling in almost 95% of cases [41]. Rapid on-site evaluation (ROSE) offers immediate feedback regarding diagnosis and quality of obtained specimens during EBUS-TBNA. The diagnostic yield with ROSE has an absolute percentage increase of 2.9–8% [37]. Studies also suggest that the use of ROSE can minimize molecular analysis failure [37]. In one study, successful molecular profiling was 10% higher in the cohort diagnosed with the addition of ROSE compared to its absence, although not statistically significant [42]. While the College of American Pathologists makes no recommendation for or against the use of one collection medium, fixative, or stain over another, it is worth noting that ethanol fixed samples with “negative” PD-L1 expression should be interpreted with caution due to a considerable rate of false negative results [37,43].

### 6.2. Endoscopic Ultrasound

Endoscopic ultrasound (EUS) fine-needle aspiration (FNA) employs the same concepts of EBUS-TBNA while using an EUS scope to obtain ultrasound-guided FNA samples via the esophagus, while EUS-B utilizes an EBUS scope. EUS-FNA/EUS-B allow access to mediastinal lymph nodes (4L, 7, 8, 9), celiac lymph nodes, the left adrenal gland, and portions of the liver, along with parenchymal lesions located close to the esophagus. The main advantages of this method, compared to EBUS-TBNA, include access to subdiaphragmatic sites. While EUS-FNA offers similar diagnostic sensitivity of 83–89%, routine use is limited only to augment EBUS-TBNA in select cases [35].

### 6.3. CT-Guided Lung Biopsy

CT-guided lung biopsy has been an important diagnostic mechanism for pulmonary nodules, most notably peripherally based lesions. Studies evaluating its diagnostic yield estimate an 82–95% accuracy [44]. Samples are obtained via FNA and/or core needle biopsy (CNB) [45,46]. CNB is preferred over FNA due to superior diagnostic and genomic analysis ability. While FNA comparatively underperforms, CNB has an impressive sensitivity and specificity for malignancy [44,45,46], with one study reporting 92% and 98.6%, respectively [47]. Compared to FNA, CNB specimens are more likely to have remaining unexhausted tissue available for genomic analysis [47,48]. While CT-guided biopsy often obtains larger samples compared to transbronchial approaches, a recently published study reveals similar overall success rates for NGS analysis between the two modalities, with CT-guided biopsy performing inferiorly to transbronchial biopsy for RNA-based NGS testing [49]. Similarly, success of PD-L1 TPS analysis was comparable between CT-guided CNB, EBUS-TBNA, and endobronchial biopsy [50].

Despite relatively high diagnostic and molecular-testing yields, CT-guided lung biopsy possesses greater complication risks compared to bronchoscopic approaches. Pneumothorax occurs in approximately 9–43% of patients; at onset, each case has a 0–15% chance of subsequent chest-tube insertion [44]. Though reportedly less common, intrapulmonary hemorrhage may occur in up to 26% of cases, with CNB representing an increased risk [44]. These tangible adverse events create motivation to develop safer diagnostic approaches with equivalent or greater yield.

### 6.4. Guided Bronchoscopy

Guided bronchoscopy allows pulmonologists access to pulmonary lesions unreachable via traditional bronchoscopy, offering an improved safety profile compared to the alternative CT-guided samplings. Brushes, forceps, and aspiration needles are all used for sampling under radial probe endobronchial ultrasound (rEBUS), electromagnetic navigational bronchoscopy (ENB), and robotic-assisted bronchoscopy (RAB) guidance. Many studies have examined the clinical efficacy of these approaches. A large meta-analysis reported a weighted diagnostic yield of 70.6% for rEBUS [51]. NAVIGATE, a large multi-institutional prospective cohort study evaluated ENB finding a 72.9% diagnostic yield for malignancy [52]. Pooled meta-analyses corroborate these results with a suggested diagnostic yield of 65–70% [53]. However, these studies did not include digital tomosynthesis-corrected (DT)-ENB. The combination of ENB and rEBUS has revealed varied outcomes for enhancing diagnostic success [52,54,55]. RAB currently operates under two platforms in the US, the Monarch system or Ion Endoluminal system. The Monarch platform employs ENB technology and was evaluated via the BENEFIT trial [56]. Here, rEBUS was used to confirm lesion localization with a 96.2% accuracy and a subsequent diagnostic yield of 74.1%; lesions with concentric view per rEBUS were associated with improved diagnostic accuracy [56]. The Ion Endoluminal system incorporates shape sensing (ss) technology, and a recent study found a 98.7% target localization rate and an overall diagnostic yield of 81.7% [57]. A retrospective investigation comparing ss-RAB and DT-ENB demonstrated comparable diagnostic yields at 77% and 80%, respectively [58]. Finally, a retrospective study comparing the diagnostic yields of RAB versus CT-guided transthoracic biopsy were 87.6% and 88.4%, respectively [59]. Respective sensitivities and specificities for malignancy were 82.1% and 100% for RAB and 88.5% and 100% for CT-guided transthoracic biopsy [59]. Advantages of guided bronchoscopy approaches include minimal complication rates, between 2–4%, with a smaller risk of serious events requiring invasive intervention or hospitalization [51,52,56,60,61]. Additionally, concurrent mediastinal staging is commonplace, along with placement of fiducial markers or pleural dye to assist in surgical and/or radiation therapy [52]. Combined, these results reveal a promising new era for pulmonary biopsy.

Equally important to diagnostic efficacy is the ability to obtain adequate tissue for comprehensive biomarker and IHC testing. An early retrospective study evaluating ENB found that 94.1% (n = 17) of specimens were viable for *EGFR* and *ALK* alteration analyses and 100% (n = 22) were acceptable for IHC analysis [62]. Most cases utilized transbronchial forceps biopsy samples for biomarker testing [62]. A more recent study reported similar results in the setting of ENB, revealing 91.7% (n = 13) of NSCLC samples were adequate for *EGFR*, *ALK,* and PD-L1 testing [63]. Transbronchial forceps instrument was also the most common sampling device used [63]. Comparing rEBUS and transbronchial biopsy to subsequent surgical specimens revealed a 3% (n = 56) false negative rate among transbronchial biopsies for *EGFR* mutational analysis, a direct result of low tumor cell quantity in the associated specimens [64]. Another recent study demonstrated tremendous success rates on genomic alteration testing with specimens obtained with transbronchial brushing under rEBUS guidance [65]. Samples were adequate in 99.1% and 85.1% with DNA and RNA sequencing, respectively [65]. While these previous studies produced promising results for limited biomarker analysis, comprehensive genomic testing has escalated along with the rapid rise in available targeted agents. One report evaluated feasibility of NGS and PD-L1 testing from samples obtained via rEBUS and ENB procedures. PD-L1 testing was successfully performed on 94% (n = 232) of cases [66]. A smaller cohort underwent subsequent surgical resection with perfect concordance among the 50% threshold (n = 15), whereas the >1% threshold identified three false negative cases (n = 8) from bronchoscopy specimens [66]. Unfortunately, exhausted tissue, leading to unacceptable NGS analysis, affected 30.9% (n = 188) of cases, disproportionately impacting early-stage disease [66]. These results suggest the importance of procuring larger tissue samples while maintaining a minimally invasive approach. A recent analysis of RAB with different sampling methods including needle aspiration, transbronchial forceps, and cryobiopsy were evaluated. Cryobiopsy produced less crush artifact and obtained a median amount of diagnostic tissue, approximately 11-fold higher, than needle aspiration and 2-fold higher than transbronchial forceps [67]. Forty-nine cases of NSCLC were analyzed via IHC, fluoresence in situ hybridization (FISH), and NGS. One hundred percent (n = 29) of cryobiopsy, 83% (n = 18) of forceps biopsy, and 50% (n = 2) of needle aspiration specimens were acceptable for such extensive analysis [67]. This study provides important insight into promising new biopsy approaches to improve concurrent diagnosis and molecular analysis.

### 6.5. Surgical Approaches

Mediastinoscopy is performed via mediastinoscope insertion through a base of neck incision. This method provides access to ipsilateral and contralateral pretracheal, paratracheal, and anterior subcarinal lymph nodes. Independently, it is generally reserved for circumstances when EBUS-TBNA/EUS-FNA fail to confirm disease in highly suspected cases [35]. A meta-analysis found the addition of mediastinoscopy following EBUS/EUS resulted in improved diagnostic sensitivity for N2/N3 disease with increased likelihood of upstaging [68]. As such, thoracic surgeons may perform mediastinoscopy prior to surgical resection.

Video-assisted thorascopic surgery (VATS) is the most common operative method used for lung cancer resection, providing access to the full tumor specimen along with a wide range of ipsilateral nodes; however, contralateral nodes are generally inaccessible [35]. Given ample attainable tissue, VATS has a 99% diagnostic sensitivity with improved ability for subsequent molecular analysis [35,69]. It can also provide information regarding pleural tumor involvement. Despite superior diagnostic abilities, VATS, as an isolated diagnostic tool, is generally reserved for select cases.

### 6.6. Thoracentesis

Patients presenting with a pleural effusion and presumed, or previously diagnosed, lung cancer should undergo a thoracentesis for at least diagnostic purposes. Malignant pleural effusions may alter disease staging and treatment recommendations. Pleural fluid has a reported 72% diagnostic yield for malignancy with 49–91% sensitivity following submission of two specimens [70]. Factors impacting diagnostic yield include sample preparation and tumor type, highly favoring adenocarcinoma histology [70,71]. Like standard tissue, biomarker testing can be performed on pleural fluid samples. High tumor cellularity, as opposed to sample amount, improves success of molecular analysis; a goal tumor cellularity of >10% is often sought [72]. Comparisons of NGS and PD-L1 testing performed on standard tissue biopsy and pleural fluid samples reveal highly congruent results of greater than 80% in multiple analyses [72]. Thoracentesis provides a minimally invasive diagnostic approach with easily attainable specimen.

### 6.7. Extra-Thoracic Disease Sampling

Lung cancer may also be diagnosed from biopsies or surgical specimens obtained from metastatic sites. Additionally, confirmatory biopsies of extrathoracic areas may be warranted to solidify metastatic disease. A 2014 study evaluated NSCLC biopsy specimens from various sites and their respective success rates for *EGFR*, *ALK,* and *KRAS* molecular/FISH analyses. While intrathoracic disease specimens comprised the majority, overall success rates reached 90–94% with small biopsies/cytologic specimens reaching 85–88% [73]. Extrathoracic site-based failure rates for *EGFR, ALK,* and *KRAS* were 17.4% (n = 23), 22.2% (n = 18), and 14.3% (n = 14) for bone; 0% (n = 23), 0% (n = 21), and 0% (n = 17) for brain; and 14.3% (n = 14), 12.5% (n = 8), and 50% (n = 4) for liver, respectively [73]. Increased failure rates were seen from nonsurgical specimens and cytologic cell block specimens from bone. Surgical specimens revealed a 0% failure rate for all mutational analyses representing the presence of ample tumor cells to undergo diagnostic and molecular testing [73]. Osseous failure relates to tissue processing and acid decalcification, which damages nucleic acids interfering with molecular analysis [73]. A more recent study assessing the role of CT-guided percutaneous osteolytic biopsy found a 100% diagnostic rate with a subsequent molecular analysis yield of 94.6% (n = 37) in the absence of tissue decalcification [74]. This underscores the importance of expert tissue handling among pathologists to maximize molecular testing.

Additionally, the adrenal glands are frequently subject to CT-guided percutaneous biopsy for diagnostic means. A large retrospective review identified the diagnostic sensitivity and specificity for lung cancer to be 97% and 100%, respectively [75]. Additionally, the procedure was remarkably safe, with only a 1.1% rate of minor, and no major, complications reported [75].

### 6.8. Liquid Biopsy

Liquid biopsy is an emerging diagnostic tool in the field of NSCLC. This noninvasive test is typically performed on plasma to identify circulating tumor cells, cell-free DNA (cfDNA), and exosomes. While tissue biopsy remains the gold standard, liquid biopsy can provide information reflecting tumor heterogeneity or metastatic disease. A major limitation of liquid biopsy is its variable sensitivity, depending on the testing platform and extent of tumor burden. Patients with extrathoracic disease have a higher likelihood of successfully detecting *EGFR* mutations on liquid biopsy compared to patients with intrathoracic metastasis [76]. Real-world studies have shown sensitivity of NGS-based cfDNA assays ranging from 75–90% [77].

As a tool for monitoring treatment response, cfDNA is emerging as a marker of minimal residual disease and prognostic information. The presence of cfDNA after curative treatment can predict disease recurrence [78,79] and may also provide valuable information to guide treatment. In one study, patients with the presence of cfDNA showed benefit from consolidation immunotherapy while patients without presence of cfDNA showed no benefit [77]. While such tests are not yet endorsed by the NCCN, it is possible that they will eventually be incorporated into SOC management of such patients.

## 7. Conclusions

Timely and accurate biomarker testing is crucial to determine optimal treatment for all stages of NSCLC. This review provides evidence regarding the utility and clinical impact of common targeted therapies and ICIs currently in use. Identification of targetable mutations and PD-L1 status not only carries direct implications for SOC treatments but can also provide important opportunities for clinical trial participation. Reflex testing to obtain all biomarker data at the time of initial diagnosis can expedite patient care, especially given high recurrence rates seen among NSCLC. At the core, pulmonologists play a central role in the diagnosis of NSCLC and ensuring all appropriate data is obtained prior to treatment planning. EBUS-TBNA and guided bronchoscopy methods are safe, effective, and minimally invasive modalities to obtain adequate specimens for diagnosis, staging, and biomarker analysis in one comprehensive procedure. As precision medicine continues to advance, it is crucial that clinicians prioritize a multidisciplinary approach in the diagnosis and treatment of NSCLC.

## Figures and Tables

**Figure 1 diagnostics-13-01117-f001:**
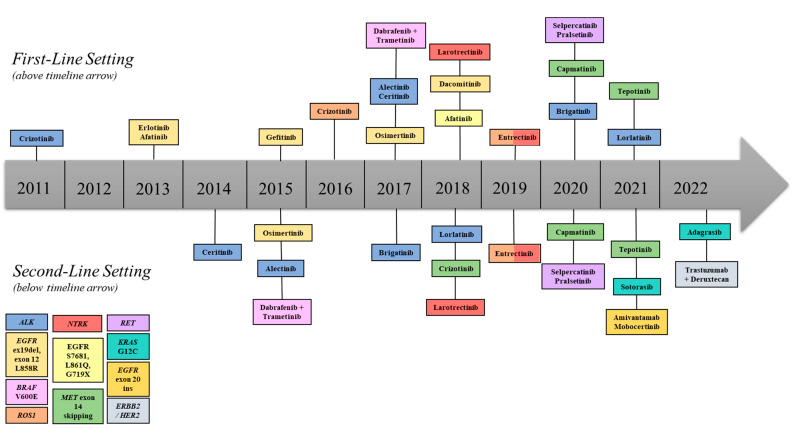
Timeline of FDA-approved targeted therapies for NSCLC. First-line therapies depicted above, and second-line therapies depicted below the central arrow.

**Figure 2 diagnostics-13-01117-f002:**
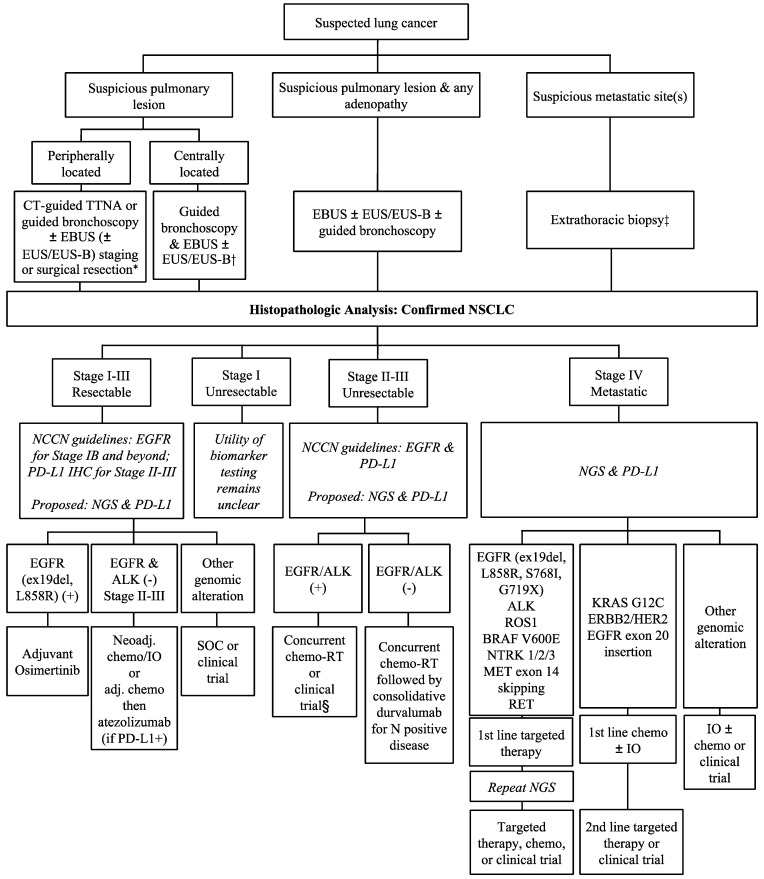
Approach to tissue biopsy and biomarker testing and treatment in NSCLC by disease stage. IO = immunotherapy; SOC = standard of care. * The utility of routine mediastinal and hilar staging for lesions < 1–2 cm is not clear at present. † Many factors need to be considered such as lung function, local expertise, patient’s preference, and probability of disease. ‡ If obvious widely metastatic disease or certainty on single/minimal metastasis on imaging, the easiest/fastest/highest yield site should be biopsied. § The benefit of consolidative durvalumab has not been established in these patients.

**Table 1 diagnostics-13-01117-t001:** Oncogenic driver mutations and respective first- and subsequent-line targeted therapies recommended by the NCCN [7,8,9]. * NCCN preferred agent.

Biomarker	Approximate Frequency iN Nsclc	First-Line Targeted Therapy	Subsequent-Line Targeted Therapy
** *EGFR* ** **Exon 19 Deletion, Exon 12 L858R Mutation**	*EGFR*: 10–20% 85% of all *EGFR* mutations	Osimertinib * Erlotinib Afatinib Gefitinib Dacomitinib Erlotinib + Ramucirumab Erlotinib + Bevacizumab (nonsquamous)	Osimertinib - In the setting of a T790M+ mutation, if not previously given
** *ALK* ** **Rearrangement**	3–7%	Alectinib * Brigatinib * Lorlatinib * Ceritinib Crizotinib	Lorlatinib * - If not previously given or in the setting of an ALK G1202R+ mutation Alectinib - Only following disease progression on Crizotinib Brigatinib- Only following disease progression on Crizotinib Ceritinib - Only following disease progression on Crizotinib
** *MET* ** **Exon 14 Skipping Mutation**	3–4%	Capmatinib * Tepotinib * Crizotinib	Capmatinib * - Given following first-line chemotherapy Tepotinib * - Given following first-line chemotherapy Crizotinib - Given following first line chemotherapy
** *EGFR* ** **S768I, L861Q, G719X**	*EGFR*: 10–20% 6% of all *EGFR* mutations	Afatinb * Osimertinib * Erlotinib Dacomitinib Gefitinib Erlotinib + Ramucirumb Erlotinib + Bevacizumab (nonsquamous)	Osimertinib - In the setting of a T790M+ mutation, if not previously given
** *ROS1* ** **Rearrangement**	1–2%	Entrectinib * Crizotinib * Ceritinib	Lorlatinib - For systemic progression, if not previously given Entrectinib - For CNS progression, if previously treated with Crizotinib or Ceritinib
** *RET* ** **Rearrangement**	1–2%	Selpercatinib * Pralsetinib * Cabozantinib	Selpercatinib * - Given following first-line chemotherapy Pralsetinib * - Given following first-line chemotherapy Cabozantinib - Given following first-line chemotherapy
** *BRAF* ** **V600E Mutation**	1–2%	Dabrafenib/Trametinib * Dabrafenib Vemurafenib	Dabrafenib/Trametinib - Given following first-line chemotherapy
** *NTRK1/2/3* ** **Gene Fusion**	<1%	Larotrectinib Entrectinib	Larotrectinib - Given following first-line chemotherapy Entrectinib - Given following first-line chemotherapy
** *KRAS* ** **G12C Mutation**	13%	N/A	Sotorasib - Given after at least first-line chemotherapy without prior KRAS G12C targeted therapy Adagrasib - Given after at least first-line chemotherapy without prior KRAS G12C targeted therapy
** *ERBB2/HER2* ** **Mutation**	2–4%	N/A	Fam-trastuzumab deruxtecan-nxki * - Given after at least first-line chemotherapy and if not previously received Ado-trastuzumab emtansine - Given after at least first-line chemotherapy and if not previously received
** *EGFR* ** **Exon 20 Insertion**	*EGFR*: 10–20% 4–10% of all *EGFR* mutations	N/A	Amivantamab-vmjw - Given after at least first-line chemotherapy and if not previously received Mobocertinib - Given after at least first-line chemotherapy and if not previously received

## Data Availability

Not applicable.

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
