# Peer review of "NSCLC in the Era of Targeted and Immunotherapy: What Every Pulmonologist Must Know"

_diagnostics, 2023, doi:10.3390/diagnostics13061117_

Round 1

Reviewer 1 Report

This review is one of plethora of similar works. It is a question if it is useful? In my opinion- no, but it depends on the decision of Editor. However, this article is well written, well illustrated and in fact, it concentrates knowledge on NSCLC from diagnosis to treatment.

I can not believe that the Authors omitted histological diagnosis of lung cancer. The recognition of subtypes of NSCLC is crucial in therapeutic decision and the choice of predictive biomarkers. Please correct this work in the aspects of current histological classification.

To summarize – in my opinion this article is not needed, all the data are widely available. The chapter on histological classification is necessary as well as  the inclusion of histological types in diagnostic algorithm. Please take into account that the review is dedicated also outside US.  

Author Response

Reviewer 1

This review is one of plethora of similar works. It is a question if it is useful? In my opinion- no, but it depends on the decision of Editor. However, this article is well written, well-illustrated and in fact, it concentrates knowledge on NSCLC from diagnosis to treatment.

I cannot believe that the Authors omitted histological diagnosis of lung cancer. The recognition of subtypes of NSCLC is crucial in therapeutic decision and the choice of predictive biomarkers. Please correct this work in the aspects of current histological classification.

To summarize – in my opinion this article is not needed, all the data are widely available. The chapter on histological classification is necessary as well as the inclusion of histological types in diagnostic algorithm. Please take into account that the review is dedicated also outside US. 

Thank you so very much for taking the time to review our manuscript. We very much appreciate your comments and think these are extremely valuable.

We respectfully disagree on the perceived lack of necessity of articles such as this. Our article has some unique features. We have created a novel algorithm that helps the reader on their decision-making process on biomarker testing based on disease stage. Second, we combine background/importance of biomarker testing and their respective therapies along with a concise review of the many technologies available to obtain tissue necessary for adequate testing. To our knowledge, most (possibly all) review articles on these subjects are not condensed into a single article that may facilitate the reader on putting two and two together. In fact, the knowledge and the quality of biomarker testing in NSCLC around the world remains underused as shown below.

  • A survey of US pulmonologists demonstrated high EGFR and ALK testing, 99% and 95%, respectively. However, routine testing of BRAF (44%), ROS1 (48%), PD-L1 (77%), HER2 (33%), KRAS (68%), and RET (15%) were quite disappointing (Fox et al. CHEST 2021; 160(6):2293-2303).
  • A US claims data investigation for years 2017-2020 on almost 15,000 advanced/metastatic NSCLC patients demonstrated a dismal 31.5% NGS testing before 1st line therapy in whites and 25.8% in black patients. Even for non-squamous NSCLC, NGS testing before 1st line therapy was very poor at 36.6% and 29.7% for blacks and whites, respectively (Bruno et al. JCO Precis Oncol, 2022 Jun;6:e2100427. doi: 10.1200/PO.21.00427).
  • The US Oncology Network (a large community-based oncology network) reported their real-world data on biomarker testing on 3474 patients with metastatic NSCLC. Only 46% received all 5 biomarker tests that were analyzed. Quoting their analysis, “testing rates from 2018 to 2020 were 71% to 71% for EGFR, 71% to 70% for ALK, 69% to 67% for ROS1, 51% to 59% for BRAF, 82% to 84% for PD-L1, and 42% to 49% for all 5 biomarkers. NGS testing increased from 33% to 45% (p < 0.0001).” These are yet again dismal biomarker testing rates (Robert et al. Lung Cancer, 2022;166:197-204).
  • Again, a large US database study of metastatic NSCLC with over 30,000 patients evaluated biomarker trends between 2015 and 2021.  By 2021, only 60% of patients had all six biomarkers (EGFR, ALK, KRAS, BRAF, ROS1, PD-L1) tested before first-line treatment (Hess et al. JTO Clin Res Rep, 2022 May 7;3(6):100336).
  • These poor testing rate numbers are not limited to the US. A small Japanese multicenter study (N=202) looking at their 2017 data demonstrated over 20% of patients with advanced/metastatic non-squamous NSCLC not having their full biomarker testing (EGFR, ALK, ROS1 and PD-L1) before first line therapy. And while they reported a very good EGFR and ALK testing, PD-L1 and ROS1 testing were suboptimal. And a number of patients did not receive appropriate treatment despite a target mutation or PD-L1>50% without such target (Shimizu et al. Ther Adv Med Oncol, 2020 Feb 22;12:1758835920904522). 
  • An ongoing German registry involving over 150 “certified lung cancer centers, comprehensive cancer centers, hospitals and office-based oncology practices located all over Germany” recently published their experience on 3,717 patients with NSCLC patients with advanced or metastatic disease between 2015 and 2019. For the year 2019, “the testing rate in patients with non-squamous NSCLC was 65.4 % for EGFR and 72.8 % for ALK.” Testing for other potential targets were even worse (Griesinger et al. Lung Cancer 2021;152:174-184).
  • A recent study evaluating the quality of biomarker testing in four Latin America countries showed data similar to the above. Quoting their article, “on an average, the frequency of molecular testing was 66%, with Argentina having the highest testing rate (79%) with a population belonging primarily to the private sector (87%) and Uruguay having the lowest testing rate (28%) but with a population belonging primarily to the public sector (91%).” In addition, presence of a genomic alteration with an available specific therapy did not necessarily mean all patients were properly treated (Martin et al. Mol Clin Oncol 2022;16(1):6).
  • In Spain, where a very robust Lung Cancer Biomarker Testing Registry (LungPath) exists, we see much of the same. Of 12,904 samples, 9,118 (71.4%) could be finally assessed. Please note, many of the not assessed samples on their analysis failed to have material for testing. It is unclear how often the samples that could be evaluated were repeat biopsies. In any event, the rate of biomarker testing was the following: EGFR (91.4%), ALK (80.1%), ROS1 (58.1%) and PD-L1 (56.2%). Other markers were not evaluated for this period (Salas C et al. J Clin Pathol 2022;75:193–200).

Based on the disappointing biomarker testing rates around the world, we strongly believe this article is very likely to expose additional readers on the crucial necessity of biomarker testing in non-squamous and even squamous NSCLC patients as clinically indicated.

Regarding tumor histology, we believe our manuscript shows clearly that genomic alteration determines specific targeted therapy irrespective of NSCLC subtype. Moreover, US national guidelines have moved away from recommending comprehensive biomarker testing only for non-squamous histology. And the reasons for that are stated in the manuscript. However, we highlight the fact that histology still remains important, especially to help and determine the type of cytotoxic therapy used in conjunction to immunotherapy for patients in whom an actionable genomic alteration is not identified. Please see the edited “Indications for Biomarker Testing” section below (the tracked modifications can be seen on the revised manuscript):

As evidenced above, appropriate biomarker and PD-L1 testing is imperative to provide optimal treatment regimens for patients with NSCLC. Comprehensive testing should be implemented regardless of tumor histology. Traditionally, testing all non-squamous histologies was advised, however squamous cell carcinomas should also be considered. Approximately 5-10% of these tumors will harbor a targetable mutation; further, tumor heterogeneity cannot be ruled out[15]. While individual tumor histology (e.g., adenocarcinoma or squamous cell carcinoma) should not be the factor to preclude or promote biomarker testing, it is particularly helpful to guide systemic therapy in addition to immunotherapy in the absence of genomic alterations as the effectiveness of different cytotoxic chemotherapy agents may be affected by intrinsic mechanisms of resistance inherent to different subtypes of NSCLC [29]. In patients with metastatic NSCLC with no actionable genomic alterations, low PD-L1 expression and/or high tumor burden cytotoxic chemotherapy remains the cornerstone of systemic therapy. Moreover, FDA-approved regimens containing the anti-angiogenic agent bevacizumab are contraindicated in patients with squamous NSCLC in view of an increased risk of high rates of life-threatening hemoptysis[30].

When considering disease stage, the NCCN recommends reflex PD-L1 IHC analysis for all locally-advanced and metastatic disease. Additionally, patients with stage II-IIIA NSCLC who undergo surgical resection and are candidates for adjuvant chemotherapy should also be tested for PD-L1 to determine eligibility for adjuvant atezolizumab. Similarly, genomic testing is recommended for all metastatic disease, and at minimum, testing for EGFR mutations is recommended among stage I-III disease[7].Arguably, comprehensive genomic and PD-L1 testing should be implemented regardless of stage, immediately at the time of diagnosis. Frequently, tissue biopsy is obtained prior to staging confirmation, and automatic reflex testing can expeditiously guide therapy once definitive stage is determined. A pragmatic example is the patient with early-stage disease that may be a candidate for neoadjuvant chemoimmunotherapy, provided the tumor does not harbor EGFR or ALK alterations. Additionally, many patients with early and locally-advanced disease will progress, necessitating this tumor-specific information. If previously obtained, this prevents risks associated with additional invasive procedures and saves valuable time, money and resources at an important treatment juncture. Furthermore, ongoing clinical trials continue to evaluate the efficacy of numerous targeted therapies in the adjuvant and neoadjuvant arenas which will likely showcase promising results leading to new FDA approvals. Ultimately, understanding a tumor’s genomic profile can expedite the implementation of late-breaking effective therapies.

Finally, not only is comprehensive genomic testing important when utilizing SOC treatments, but it is equally necessary when considering clinical trial participation, with enrollment eligibility often hinging upon these results. Unsurprisingly, trials involving targeted therapies require this information, as do those involving immunotherapies, as certain genomic alterations are known to augment ICI effectiveness. Assuming each patient is a clinical trial candidate at the time of diagnosis will save time and more importantly the risk of an additional future procedure.

We hope the comments and data herewith will address your concerns.

Thank you again for your insights.

Reviewer 2 Report

The manuscript provides some information about lung cancer biological therapy (EGFR, ALK) and immunotherapy. There is also some information on biopsy techniques for lung cancer.

It is not clear why the title of the manuscript indicates that it contains specific information specifically for pulmonologists. In many countries, pulmonologists not only diagnose and stage lung cancer but are also very familiar with chemotherapy, biological therapy, and immunotherapy and prescribe systemic treatment for lung cancer.

The information presented in the manuscript is well-known to specialists, including pulmonologists, and especially pulmonologists. Many published articles provide much more detailed reviews of both biopsy techniques and lung cancer systemic therapies. The submitted manuscript is not unique or new in any way.

Author Response

Reviewer 2

The manuscript provides some information about lung cancer biological therapy (EGFR, ALK) and immunotherapy. There is also some information on biopsy techniques for lung cancer.

It is not clear why the title of the manuscript indicates that it contains specific information specifically for pulmonologists. In many countries, pulmonologists not only diagnose and stage lung cancer but are also very familiar with chemotherapy, biological therapy, and immunotherapy and prescribe systemic treatment for lung cancer.

The information presented in the manuscript is well-known to specialists, including pulmonologists, and especially pulmonologists. Many published articles provide much more detailed reviews of both biopsy techniques and lung cancer systemic therapies. The submitted manuscript is not unique or new in any way.

Thank you so very much for taking the time to review our manuscript. We very much appreciate your comments and think these are extremely valuable.

We chose this specific title because it is our understanding this is a special edition of Diagnostics in the area of interventional pulmonology/pulmonology. But if you and the editors believe this article may have a broader audience, which we believe it does, we would be happy to change the article title to NSCLC in the Era of Targeted and Immunotherapy: What Every Non-Oncologist Must Know.”

We respectfully disagree on the perceived lack of necessity of articles such as this. Our article has some unique features. We have created a novel algorithm that helps the reader on their decision-making process on biomarker testing based on disease stage. Second, we combine background/importance of biomarker testing and their respective therapies along with a concise review of the many technologies available to obtain tissue necessary for adequate testing. To our knowledge, most (possibly all) review articles on these subjects are not condensed into a single article that may facilitate the reader on putting two and two together. In fact, the knowledge and the quality of biomarker testing in NSCLC around the world remains underused as shown below.

  • A survey of US pulmonologists demonstrated high EGFR and ALK testing, 99% and 95%, respectively. However, routine testing of BRAF (44%), ROS1 (48%), PD-L1 (77%), HER2 (33%), KRAS (68%), and RET (15%) were quite disappointing (Fox et al. CHEST 2021; 160(6):2293-2303).
  • A US claims data investigation for years 2017-2020 on almost 15,000 advanced/metastatic NSCLC patients demonstrated a dismal 31.5% NGS testing before 1st line therapy in whites and 25.8% in black patients. Even for non-squamous NSCLC, NGS testing before 1st line therapy was very poor at 36.6% and 29.7% for blacks and whites, respectively (Bruno et al. JCO Precis Oncol, 2022 Jun;6:e2100427. doi: 10.1200/PO.21.00427).
  • The US Oncology Network (a large community-based oncology network) reported their real-world data on biomarker testing on 3474 patients with metastatic NSCLC. Only 46% received all 5 biomarker tests that were analyzed. Quoting their analysis, “testing rates from 2018 to 2020 were 71% to 71% for EGFR, 71% to 70% for ALK, 69% to 67% for ROS1, 51% to 59% for BRAF, 82% to 84% for PD-L1, and 42% to 49% for all 5 biomarkers. NGS testing increased from 33% to 45% (p < 0.0001).” These are yet again dismal biomarker testing rates (Robert et al. Lung Cancer, 2022;166:197-204).
  • Again, a large US database study of metastatic NSCLC with over 30,000 patients evaluated biomarker trends between 2015 and 2021.  By 2021, only 60% of patients had all six biomarkers (EGFR, ALK, KRAS, BRAF, ROS1, PD-L1) tested before first-line treatment (Hess et al. JTO Clin Res Rep, 2022 May 7;3(6):100336).
  • These poor testing rate numbers are not limited to the US. A small Japanese multicenter study (N=202) looking at their 2017 data demonstrated over 20% of patients with advanced/metastatic non-squamous NSCLC not having their full biomarker testing (EGFR, ALK, ROS1 and PD-L1) before first line therapy. And while they reported a very good EGFR and ALK testing, PD-L1 and ROS1 testing were suboptimal. And a number of patients did not receive appropriate treatment despite a target mutation or PD-L1>50% without such target (Shimizu et al. Ther Adv Med Oncol, 2020 Feb 22;12:1758835920904522). 
  • An ongoing German registry involving over 150 “certified lung cancer centers, comprehensive cancer centers, hospitals and office-based oncology practices located all over Germany” recently published their experience on 3,717 patients with NSCLC patients with advanced or metastatic disease between 2015 and 2019. For the year 2019, “the testing rate in patients with non-squamous NSCLC was 65.4 % for EGFR and 72.8 % for ALK.” Testing for other potential targets were even worse (Griesinger et al. Lung Cancer 2021;152:174-184).
  • A recent study evaluating the quality of biomarker testing in four Latin America countries showed data similar to the above. Quoting their article, “on an average, the frequency of molecular testing was 66%, with Argentina having the highest testing rate (79%) with a population belonging primarily to the private sector (87%) and Uruguay having the lowest testing rate (28%) but with a population belonging primarily to the public sector (91%).” In addition, presence of a genomic alteration with an available specific therapy did not necessarily mean all patients were properly treated (Martin et al. Mol Clin Oncol 2022;16(1):6).
  • In Spain, where a very robust Lung Cancer Biomarker Testing Registry (LungPath) exists, we see much of the same. Of 12,904 samples, 9,118 (71.4%) could be finally assessed. Please note, many of the not assessed samples on their analysis failed to have material for testing. It is unclear how often the samples that could be evaluated were repeat biopsies. In any event, the rate of biomarker testing was the following: EGFR (91.4%), ALK (80.1%), ROS1 (58.1%) and PD-L1 (56.2%). Other markers were not evaluated for this period (Salas C et al. J Clin Pathol 2022;75:193–200).

Based on the disappointing biomarker testing rates around the world, we strongly believe this article is very likely to expose additional readers on the crucial necessity of biomarker testing in non-squamous and even squamous NSCLC patients as clinically indicated.

We also do understand there are pulmonologists around the world that are also the same physician who prescribe treatment for patients with lung cancer. However, as multinational authors with broad knowledge on how diagnosis and treatment of lung cancer patients functions in a many countries, we do not believe this to be a typical scenario. Therefore, we expect this article to be quite informative to a large number of non-oncologists. 

We hope the comments and data herewith will address your concerns.

Thank you again for your insights.

Reviewer 3 Report

Thank you for giving me the opportunity to review this manuscript.

The authors reviewed molecular targeted therapy, immune therapy, and the biomarker testing for patients with NSCLC.

This review is well written overall, but some minor revisions are needed in the current manuscript. Please consider the followings.

Minor comment:
1. 6.4. Guided bronchoscopy

There is a pivotal prospective cohort study which investigated suitability of cytological specimens obtained by bronchoscopy with using EBUS-guide-sheath (EBUS-GS) for peripheral lung leasions.

You should cite following article.

Furuya N, et al. Suitability of transbronchial brushing cytology specimens for next-generation sequencing in peripheral lung cancer. Cancer Sci. 2021 Jan;112(1):380-387.

Author Response

Reviewer 3

Thank you for giving me the opportunity to review this manuscript.

The authors reviewed molecular targeted therapy, immune therapy, and the biomarker testing for patients with NSCLC.

This review is well written overall, but some minor revisions are needed in the current manuscript. Please consider the followings.

Minor comment:
1. 6.4. Guided bronchoscopy

There is a pivotal prospective cohort study which investigated suitability of cytological specimens obtained by bronchoscopy with using EBUS-guide-sheath (EBUS-GS) for peripheral lung leasions.

You should cite following article.

Furuya N, et al. Suitability of transbronchial brushing cytology specimens for next-generation sequencing in peripheral lung cancer. Cancer Sci. 2021 Jan;112(1):380-387.

Thank you so very much for taking the time to review our manuscript. We very much appreciate your comments and suggestion.

The paper you suggested does indeed show impressive results for transbronchial brushings. We have added the following to the guide bronchoscopy section: “Another recent study demonstrated tremendous success rates on genomic alteration testing with specimens obtained with transbronchial brushing under rEBUS guidance.(Furuya N et al citation here.) Samples were adequate in 99.1% and 85.1% with DNA and RNA sequencing, respectively.”

We hope the comments and data herewith will address your concerns.

Thank you again for your insights.

Round 2

Reviewer 1 Report

Thank to the Authors for the explanation. It is evidently US point of vue and our discussion could go on forever. I appreciate the effort and accept your point of view.

Reviewer 2 Report

None.